# SyncVIS: Synchronized Video Instance Segmentation

**Rongkun Zheng**[1]    **Lu Qi**[2]    **Xi Chen**[1]    **Yi Wang**[3]
**Kun Wang**[4]    **Yu Qiao**[3]    **Hengshuang Zhao**[1*]
[1]The University of Hong Kong [2]University of California, Merced
[3]Shanghai Artificial Intelligence Laboratory [4]SenseTime Research
{zrk22@connect, hszhao@cs}.hku.hk

## Abstract

Recent DETR-based methods have advanced the development of Video Instance Segmentation (VIS) through transformers' efficiency and capability in modeling spatial and temporal information. Despite harvesting remarkable progress, existing works follow asynchronous designs, which model video sequences via either video-level queries only or adopting query-sensitive cascade structures, resulting in difficulties when handling complex and challenging video scenarios. In this work, we analyze the cause of this phenomenon and the limitations of the current solutions, and propose to conduct synchronized modeling via a new framework named SyncVIS. Specifically, SyncVIS explicitly introduces video-level query embeddings and designs two key modules to synchronize video-level query with frame-level query embeddings: a synchronized video-frame modeling paradigm and a synchronized embedding optimization strategy. The former attempts to promote the mutual learning of frame- and video-level embeddings with each other and the latter divides large video sequences into small clips for easier optimization. Extensive experimental evaluations are conducted on the challenging YouTube-VIS 2019 & 2021 & 2022, and OVIS benchmarks, and SyncVIS achieves state-of-the-art results, which demonstrates the effectiveness and generality of the proposed approach. The code is available at https://github.com/rkzheng99/SyncVIS.

## 1   Introduction

Video Instance Segmentation (VIS) is a fundamental while challenging vision task that aims to detect, segment, and track object instances inside videos based on a set of predefined object categories at the same time. With the prosperous video media, VIS has attracted various attention due to its numerous vital applications in areas such as video understanding, video editing, autonomous driving, etc.

Benefiting from favorable long-range modeling among frames, query-based offline VIS methods [6, 31, 15, 29, 17, 36] like Mask2Former-VIS [6], and SeqFormer [31] begin to dominate the VIS. Inspired by the object detection method DETR [5], they learn a group of queries that can track and segment potential instances simultaneously across the multiple frames of a video. On the other hand, online VIS approaches like IDOL [32] also exploit the temporal consistency of query embeddings and associate instances via linking the corresponding query embeddings frame by frame. Albeit the success gained by those methods, we find they barely capitalize multi-frame inputs. In practice, the Mask2Former-VIS [6] would significantly perform worse if more input frames are given during training (evidenced in Fig. 3). This is paradoxical to our common sense that more frames could facilitate deep learning models obtaining more motion information of instances.

For this problem, many researchers [15, 31, 17] point out that the video-level queries are vitally hard to track the instances well if receiving many frames in training. That is because the trajectory

---

[*]Corresponding author

38th Conference on Neural Information Processing Systems (NeurIPS 2024).

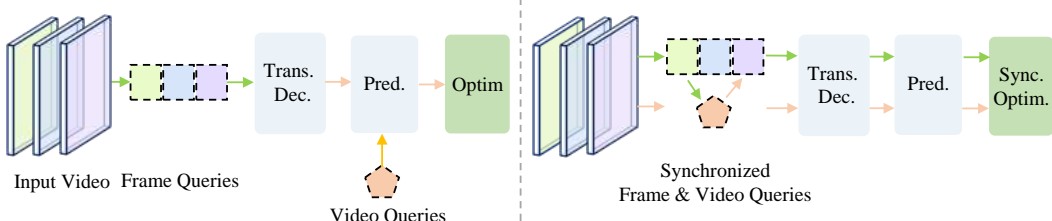

Fig. 1. Comparison of video instance segmentation paradigms. Previous methods (left part) like VITA [15] adopt **asynchronous** query-sensitive structures to model instance appearances and trajectories. Our model (right part) employs frame and video embeddings in a query-robust **synchronous** manner, and they synchronize with each other through the transformer decoder to generate the refined video-level query embeddings for the prediction. Also, we employ a synchronized embedding optimization strategy 'Sync. Optim.' instead of the classic optimization approach.

complexity will increase in polynomials along with the number of frames. Therefore, state-of-the-art methods like SeqFormer [31] and VITA [15] usually decouple the trajectory into spatial and temporal dimensions, which are modeled by frame-level and video-level queries, respectively. Specifically, they utilize the frame-level queries to segment each frame independently and then associate these frame-level queries with video-level queries, which are responsible for the final video-level prediction. The well-trained frame-level queries guarantee the quality in the spatial dimension and thus decrease the burden of video queries. However, we argue that two issues remain in these asynchronous designs (as illustrated at the left of Fig. 1). First, with the asynchronous structure, the wellness of video-level queries heavily relies on the learning of former frame-level queries, inside which some motion information may be lost because it is an image encoding stage (rather than video encoding), which leads to the sensitivity of queries to the learning quality of former stages. Second, previous works have not solved the bipartite matching among more frames (rather than single frame), and thus the optimization complexity of trajectories remains exorbitant. Both two issues block the further development of query-based methods for video instance segmentation.

To this end, we propose to model video and frame queries synchronously with a new framework named SyncVIS to address the above-mentioned issues. Built upon DETR-style structures [6, 32], our SyncVIS has two key components: the synchronized video-frame modeling paradigm and the synchronized embedding optimization strategy. Both designs put effort into unifying the frame- and video-level predictions in synchronization. The synchronized video-frame modeling paradigm makes frame- and video-level embeddings interact with each other in a query-robust parallel manner, rather than a query-sensitive cascade structure. Then the synchronized embedding optimization strategy adds a video-level buffer state to generate more tractable intermediate bipartite matching optimization compared with only frame-level losses. Fig. 1 demonstrates the schematic difference between the asynchronous state-of-the-art method and our synchronous approach. Our model is schematically simple but practically more effective, with exquisite designs as follows.

In the synchronized video-frame modeling paradigm, we employ frame and video-level embeddings in the transformer decoder to model object segmentation and tracking synchronously. Specifically, frame-level embeddings are assigned to each sampled frame, and responsible for modeling the appearance of instances, and video-level embeddings are a set of shared instance queries for all sampled frames, which are used to characterize the general motion (In the DETR-style architecture, when video queries are associated with features across time via the decoder, they can effectively model instance-level motion through the cascade structure. In Mask2Former-VIS, the use of video queries alone enables the capture of instance motion). Frame-level embeddings are kept on each frame to attend to instances locally. In each decoder layer, the video-level embeddings are aggregated to refine frame-level embeddings on the corresponding frame. The refined frame-level embeddings, in turn, are aggregated into video-level embeddings. By repeating this synchronization in decoder layers, SyncVIS incorporates the semantics and movement of instances in each frame. In the synchronized embedding optimization strategy, we focus more on video-level bipartite matching. Concretely, we decouple the input video into several clips to synchronize video and frame, and the total number of clips is related to the combinatorial number. Then, we calculate each clip loss independently by video-level bipartite matching, so that video embeddings can maintain their association ability.

We evaluate our SyncVIS on four popular VIS benchmarks, including YouTube-VIS 2019 & 2021 & 2022 [34], and OVIS-2021 [27]. The experiments show the effectiveness of our method with signifi-

cant improvement over the current state-of-the-art methods VITA [15], DVIS [38], and CTVIS [37]. Our contributions are as follows:

- We analyze the limitations of existing video instance segmentation methods and propose a framework named SyncVIS with synchronized video-frame modeling. It can well characterize instances' trajectories under complex and challenging video scenarios.
- We develop two critical modules: a synchronized video-frame modeling paradigm and a synchronized embedding optimization strategy. The former adopts a synchronized paradigm to alleviate error accumulation in cascade structures. The latter divides large video sequences into small clips for easier optimization.
- We conduct extensive experimental evaluations on challenging VIS benchmarks, including YouTube-VIS 2019 & 2021 &2022, and OVIS 2021, and the achieved state-of-the-art results demonstrate the effectiveness and generality of the proposed approach.

## 2 Related Works

**Online video instance segmentation.** Most online VIS methods adopt the tracking-by-detection paradigm, integrating a tracking branch into image instance segmentation models. These methods predict detection and segmentation within a local range using a few frames and associate these outputs using matching algorithms. MaskTrack R-CNN [34] incorporates a tracking branch to Mask R-CNN [12]. Many subsequent approaches [4, 35, 21], follow this pipeline, measuring the similarities between frame-level predictions and associating them with different matching modules. CrossVIS [35] uses the instance feature in the current frame to pixel-wisely localize the same instance in another frame. MinVIS [16] implements a query-based image instance segmentation model [7] on individual frames and associate query embeddings via bipartite matching.

Contrarily, some previous works [9, 19, 10, 14], draw inspiration from Video Object Segmentation [25], Multi-Object Tracking [8, 24, 41, 2, 26, 39], and Multi-Object Tracking and Segmentation [28]. GenVIS [14] adopts a novel target label assignment strategy and builds instance prototype memory in query-based sequential learning. IDOL [32], based on Deformable-DETR [42], introduces a contrastive learning head that acquires discriminative instance embeddings for association [11]. CTVIS [37] improved upon IDOL by constructing a consistent paradigm for both training and inference. However, online VIS methods usually adopt frame-level query and ignore the video-level associations across non-adjacent frames, which is problematic when handling complex long videos.

**Offline video instance segmentation.** Offline methods predict instance masks and trajectories through the whole video in one step using the whole video as input. STEm-Seg [1] proposes a single-stage model which learns and clusters the spatio-temporal embeddings. MaskProp [3] and Propose-Reduce [19] improve association and mask quality by mask propagation. Efficient-VIS [30] uses a tracklet query paired with a tracklet proposal to represent object instances. VisTR [29] successfully adapts DETR [5] to VIS, using instance queries to model the whole video. IFC [17] proposes inter-frame communication transformers, using memory tokens to model associations across frames. By adapting Mask2Former [7] to 3D spatio-temporal features, Mask2Former-VIS [6] becomes the state-of-the-art by exploiting its mask-oriented representation. TeViT [36] introduces a new approach based on transformers instead of the CNN backbone and associates temporal information efficiently. SeqFormer [31] decomposes the shared instance queries into frame-level box ones and utilizes video-level instance queries to relate different frames. Recently, VITA [15] uses object tokens to represent the whole video and employs video queries to decode semantics from object tokens. TMT-VIS [40] manages to jointly train multiple datasets to improve performance via different taxonomy information. However, these methods typically implement only video query or utilize asynchronous structures, and the final query-sensitive approaches have difficulties dealing with complex scenarios.

## 3 Method

Video instance segmentation can be formulated into a set prediction problem, which can be addressed by a DETR [5] style framework like Mask2Former [7]. We first revisit the Mask2Former-VIS [6], one of the baselines that our method is built on. Then we propose a synchronized transformer framework named SyncVIS to address challenging video scenarios, with its two key designs.

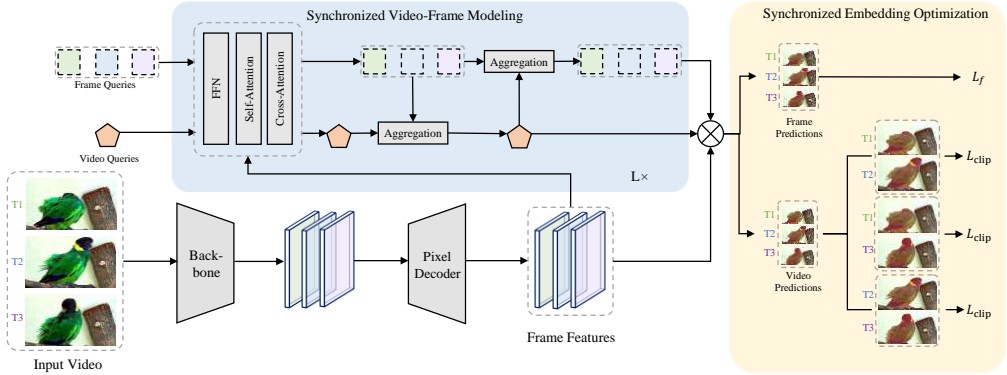

Fig. 2. Overview of the proposed synchronous video-frame modeling framework SyncVIS. The developed synchronized video-frame modeling paradigm enables video-level embeddings and frame-level ones to synchronize with each other in each stage of the decoder. SyncVIS also suggests a new synchronized embedding optimization strategy. As shown in the right part, SyncVIS decouples the input video frames into several sub-clips and feeds each sub-clip into the mask and classification head. By applying these modules, SyncVIS can incorporate both semantics and movement of instances in each frame in a synchronous manner for superior characterizing ability.

## 3.1 Revisiting Mask2Former

Mask2Former [6, 7] is a universal Transformer-based framework for image or video instance segmentation. Given an input sample, Mask2Former adopts a Transformer-based decoder, which first learns $N$ number of $C$-dimensional queries $\mathbf{Q} \in \mathbb{R}^{N \times C}$ to generate embeddings $\mathbf{E} \in \mathbb{R}^{N \times 1 \times 1 \times 1 \times C}$, then predicts $N$ segmentation masks based on the generated embeddings, where the 2nd, 3rd, and 4th dimensions of $\mathbf{E}$ correspond to temporal $T$, height $H$, and width $W$ dimensions respectively. Here, we note that the transformer decoder is a nine-layer structure, where $l$th layer cascades a masked cross-attention $h_{\mathrm{CA}}^{l}$, a self-attention $h_{\mathrm{SA}}^{l}$, and a feed-forward network FFN$^{l}$. For frame-level Mask2Former, $\mathbf{E}$ is expanded along spatial dimensions $W$ and $H$ to the shape of $N \times 1 \times H \times W \times C$. Alternatively, for video-level Mask2Former, $\mathbf{E}$ is with a shape of $N \times T \times H \times W \times C$, and the combination of temporal $T$ and spatial dimensions $W$ and $H$ enables Mask2Former to utilize the shared embeddings to represent the same visual instances across different frames consistently. Finally, $\mathbf{E}$ is utilized for instance-level classification $\mathbf{P^c}$ and pixel-level mask prediction $\mathbf{P^m}$.

**Analysis.** Although Mask2Former-VIS [6] has achieved impressive results, it exhibits notable performance degradation when dealing with complex videos. For instance, we observe a decrease in average precision (AP) of 1.5% when the number of input frames increases to ten. This observation is counter-intuitive as we expect models to improve their performance with an increased number of training frames. We hypothesize that this decline in performance stems from the insufficiency of stand-alone video queries for effective long-range video modeling. In the case of challenging long-range video sequences, there is a need to model more instances and their corresponding movements using video-level queries. This unexpected scenario suggests that there is a significant demand for distinct sets of queries that can effectively characterize both the object categories and movement trajectories in video sequences.

## 3.2 Overall Architecture

The SyncVIS is a new framework designed to improve the representation of long video frame information and optimize system learning processes. It combines video-level and frame-level embeddings synchronously, which enhances the overall functionality of the framework. The framework is depicted in Fig. 2 and features two fundamental modules, i.e., a synchronized video-frame modeling paradigm (Sec. 3.3) and a combinatorial embedding optimization strategy (Sec. 3.4).

## 3.3 Synchronized Video-Frame Modeling

Synchronized video-frame modeling is a strategy designed to avoid the sensitive cascading in previous methods and improve the synchrony between the frame-level embeddings $\mathbf{X}_{\mathrm{f}}^{l} \in \mathbb{R}^{T \times N \times C}$ and video-

level $\mathbf{X}_v^l \in \mathbb{R}^{1 \times N \times C}$. $\mathbf{X}_f^l$ focuses on every frame separately, while $\mathbf{X}_v^l$ mainly interacts with the whole video features.

Based on the design of the transformer decoder, we concurrently introduce frame- and video-level embeddings to each layer. Here, the frame- and video-level embeddings are replicated for $T$ and $1$ times by learnable frame- and video-level embeddings at first when given a video with $T$ frames. Thus both the frame- and video-level embeddings pass the transformer decoder layer and two kinds of interaction operations for synchronous exchange and refinement. For each step, these two embeddings are updated as follows:

$$\mathbf{X}_t^{l+1} = \text{FFN}^t(h_{\text{SA}}^t(h_{\text{CA}}^t(\mathbf{X}_t^l, \mathbf{F}))), \tag{1}$$

where $t \in \{f, v\}$ indicates the frame- or video-level embeddings and $\mathbf{F}$ means the pyramid features extracted from the backbone. $\mathbf{X}^l$ is the embeddings processed by the $l^{\text{th}}$ transformer decoder layer. The $h_{\text{CA}}^v(q, r)$ indicates the cross-attention with video-level query embedding $q$ and frame-level reference embedding $r$. In our design, frame-level embeddings are assigned to each sampled frame, and responsible for modeling the appearance of instances, and video-level embeddings are a set of shared instance queries for all sampled frames, which are used to characterize the general motion (because they encode the position information of instances across frames, and thereby naturally contain the motion information).

Then, we feed the frame- and video-level embeddings into the proposed synchronous structure for mutual information exchange and refinement as follows:

$$\mathbf{X}_f^{l+1} = \lambda \cdot h_{\text{CA}}^f(\mathbf{X}_f^{l+1}, \text{FFN}^{vf}(\mathbf{X}_{v\text{-s}}^{l+1})) + (1 - \lambda) \cdot \mathbf{X}_f^{l+1}, \tag{2}$$

$$\mathbf{X}_v^{l+1} = \lambda \cdot h_{\text{CA}}^v(\mathbf{X}_v^{l+1}, \text{FFN}^{fv}(\mathbf{X}_{f\text{-s}}^{l+1})) + (1 - \lambda) \cdot \mathbf{X}_v^{l+1}, \tag{3}$$

where 'v-s' and 'f-s' mean that we only select top $N_k$ embeddings in key and value to interact with the query, while 'fv' and 'vf' indicate the refinement direction of the feedfoward network, from frame to video and video to frame. $\lambda$ (set to 0.05) is the update momentum of video-level embeddings, because we presume that the aggregation of frame-level features should not change the general video-level embeddings significantly, and vice versa.

The motivation behind this approach is similar to that of the masked attention mechanism used in Mask2Former. The key difference lies in the dimension where the masking happens. In Mask2Former, the strategy is to mask out the background regions within the spatial dimension. On the other hand, our method works differently by masking out background embeddings within the key and value dimensions. This is done by selecting the top $N_k$ embeddings based on the confidence scores provided by the prediction head. Therefore, while both methods aim to reduce the influence of irrelevant background information, they do so in different ways: Mask2Former masks spatially, while our method targets key and value embeddings.

### 3.4 Synchronized Embedding Optimization

Video-level bipartite matching is a challenging memory-costly problem that remains asynchronous: The matching approaches from previous VIS methods are adapted directly from DETR, so the complexity of matching increases with the number of frames in a video, as the instances are not restrained to a single frame, but could be in any frame in the video. Even though larger input frames can bring more trajectory information of instances for prediction, this presents a challenge due to the resulting trajectory complexity, which scales polynomically with the input. Conversely, when the input is insufficient, the system may lack the necessary information to function optimally. Such asynchrony is the motivation of our new optimization strategy.

Regarding this, we present the synchronized embedding optimization strategy using the divide-and-conquer: if we want to associate frame $t_i$ to frame $t_j$ and yet the time interval may be large, an effective approach is to find $k, s.t \quad i < k < j$, and associate $t_i$ to $t_k$ as well as $t_k$ to $t_j$. When the model achieves better segmentation results on sub-clips, combining these local optimums and we can achieve a better matching. Therefore, when generating the output predictions, we would divide the predictions into several sub-clips, and optimize each sub-clips independently. This sub-clip is like a video-level buffer to help synchronizing video-level and frame-level embeddings. By optimizing the local sub-sequence of the video, rather than the entire video sequence, if the target instance becomes occluded in certain frames, our optimizing strategy can adjust the features within the sub-sequence to adapt to this change, without being affected by the unoccluded frames. The size of sub-clips, $T_s$, is variable across all VIS datasets: as for VIS datasets with fewer instances per video, such as

Youtube-VIS 2019, $T_s$ is set to 3, while for OVIS, $T_s$ works best at 2 (discussed in Sec. 4.4). In order to further reduce the complexity for better optimization, dividing into smaller sub-clips can accelerate the optimization. Also, keeping the size of two is able to maintain the temporal information. In this way, our video-level objective $\mathcal{L}_\mathbf{v}$ could be divided into several clips as follows:

$$\mathcal{L}_\mathbf{v} = \sum_{0 \le i \ne j \le T} \mathcal{L}_{\mathbf{clip(i,j)}}, \tag{4}$$

where $T$ indicates the number of input frames. And the overall training loss $\mathcal{L}$ for our model can be formulated as:

$$\mathcal{L} = \sum_{\mathbf{k} \in \{\mathbf{v,f}\}} \mathcal{L}_\mathbf{k}^{\mathrm{ce}}(\mathbf{P_k^c}, \mathbf{G_k^c}) + \sum_{\mathbf{k} \in \{\mathbf{v,f}\}} \mathcal{L}_\mathbf{k}^{\mathrm{bce}}(\mathbf{P_k^m}, \mathbf{G_k^m}) +$$
$$\sum_{\mathbf{k} \in \{\mathbf{v,f}\}} \mathcal{L}_\mathbf{k}^{\mathrm{dice}}(\mathbf{P_k^m}, \mathbf{G_k^m}) + \mathcal{L}_{\mathbf{contras}}, \tag{5}$$

where $\mathcal{L}_\mathbf{f}^{\mathrm{ce}}$ and $\mathcal{L}_\mathbf{v}^{\mathrm{ce}}$ denote the cross-entropy loss for frame- and video-level classification. Similarly, $\mathcal{L}_\mathbf{f}^{\mathrm{bce}}$, $\mathcal{L}_\mathbf{v}^{\mathrm{bce}}$, $\mathcal{L}_\mathbf{f}^{\mathrm{dice}}$, and $\mathcal{L}_\mathbf{v}^{\mathrm{dice}}$ denote the binary cross-entropy and dice loss for frame- and video-level mask prediction, respectively. Here $\mathbf{P}$ is the prediction, and $\mathbf{G}$ is the ground truth, and $\mathbf{c}$ refers to classification while $\mathbf{m}$ refers to mask. $\mathcal{L}_{\mathbf{contras}}$ represents the contrastive loss, which is applied in online settings (but not in offline) as IDOL [32] does, where the previous frame is set as a reference frame and the current frame is set as key frame.

### 3.5 Implementation Details

Our method is built on detectron2 [33]. Hyper-parameters regarding the pixel and transformer decoder are the same as these of Mask2Former-VIS [6]. In the synchronized video-frame modeling, we set the number of frame-level and video-level embeddings $N$ to 100. To extract the key information, we set the $N_k$ to 10. Following the design of Mask2Former-VIS [6], we first trained our model on COCO [20] before training on VIS datasets. We use the AdamW [23] optimizer with a base learning rate of 5e-4 on Swin-Large backbone in YoutubeVIS 2019 (we use different training iterations and learning rates for different datasets). During inference, each frame's shorter side is resized to 360 pixels for ResNet [13] and 448 pixels for Swin [22]. Most of our experiments are conducted on 4 A100 GPUs (80G), and on a cuda 11.1, PyTorch 3.9 environment. The training time is approximately 1.5 days when training with the Swin-L backbone.

## 4 Experiments

**Datasets and metrics.** YouTube-VIS dataset is a large-scale video database for video instance segmentation. The dataset has seen three iterations, in 2019, 2021, and 2022, with each adding more challenges to the dataset [34]. The first iteration, YouTube-VIS 2019, contains 2.9k videos with an average duration of 4.61 seconds. The validation set has an average length of 27.4 frames per video and covers 40 predefined categories. The dataset was updated to YouTube-VIS 2021 with longer videos with more complex trajectories. As a result, the validation videos' average length increased to 39.7 frames. The most recent update, YouTube-VIS 2022, adds an additional 71 long videos to the validation set and 89 extra long videos to the test set.

OVIS dataset is another resource for video instance segmentation, particularly focusing on scenarios with severe occlusions between objects [27]. It consists of 25 object categories and 607 training videos. Despite a smaller number of training videos compared to the YouTube-VIS datasets, the OVIS videos are much longer, averaging 12.77 seconds each. OVIS emphasizes the complexity of the scenes and the severity of occlusions between objects.

### 4.1 Main Results

We compare SyncVIS with state-of-the-art approaches which are with ResNet-50 and Swin-L backbones on the YouTube-VIS 2019 & 2021 & 2022 [34] & OVIS 2021 [27] benchmarks. The results are reported in Tables 1 , 2 and 3.

**YouTube-VIS 2019.** Table 1 shows the comparison on YouTube-VIS 2019. When applying our design to CTVIS, we discover that the forward passing of CTVIS is still asynchronous. While a

Table 1. Results comparison on the YouTube-VIS 2019 and 2021 validation sets. We group the results by online or offline methods, and then with ResNet-50 or Swin-L backbone structures. SyncVIS is the model to which we add our two designs based on CTVIS and VITA. Typically, since our design is orthogonally designed for decoder and optimization, our module could seamlessly integrate with both online & offline approaches without bells and whistles. Our algorithm gets the best AP performance under all of the settings.

| | Method | Backbone | YouTube-VIS 2019 | | | | | YouTube-VIS 2021 | | | | |
|---|---|---|---|---|---|---|---|---|---|---|---|---|
| | | | AP | $AP_{50}$ | $AP_{75}$ | $AR_1$ | $AR_{10}$ | AP | $AP_{50}$ | $AP_{75}$ | $AR_1$ | $AR_{10}$ |
| Online | CrossVIS [35] | ResNet-50 | 36.3 | 56.8 | 38.9 | 35.6 | 40.7 | 34.2 | 54.4 | 37.9 | 30.4 | 38.2 |
| | MaskTrack R-CNN [34] | ResNet-50 | 38.6 | 56.3 | 43.7 | 35.7 | 42.5 | 36.9 | 54.7 | 40.2 | 30.6 | 40.9 |
| | MinVIS [16] | ResNet-50 | 47.4 | 69.0 | 52.1 | 45.7 | 55.7 | 44.2 | 66.0 | 48.1 | 39.2 | 51.7 |
| | TCOVIS [18] | ResNet-50 | 52.3 | 73.5 | 57.6 | 49.8 | 60.2 | 49.5 | 71.2 | 53.8 | 41.3 | 55.9 |
| | IDOL [32] | ResNet-50 | 49.5 | 74.0 | 52.9 | 47.7 | 58.7 | 43.9 | 68.0 | 49.6 | 38.0 | 50.9 |
| | DVIS [38] | ResNet-50 | 51.2 | 73.8 | 57.1 | 47.2 | 59.3 | 46.4 | 68.4 | 49.6 | 39.7 | 53.5 |
| | CTVIS [37] | ResNet-50 | 55.1 | 78.2 | 59.1 | 51.9 | 63.2 | 50.1 | 73.7 | 54.7 | 41.8 | 59.5 |
| | **SyncVIS** | ResNet-50 | **57.9** | **81.3** | **60.8** | **53.1** | **64.4** | **51.9** | **74.3** | **56.3** | **43.0** | **60.4** |
| | MinVIS [16] | Swin-L | 61.6 | 83.3 | 68.6 | 54.8 | 66.6 | 55.3 | 76.6 | 62.0 | 45.9 | 60.8 |
| | DVIS [38] | Swin-L | 63.9 | 87.2 | 70.4 | 56.2 | 69.0 | 58.7 | 80.4 | 66.6 | 47.5 | 64.6 |
| | TCOVIS [18] | Swin-L | 64.1 | 86.6 | 69.5 | 55.8 | 69.0 | 61.3 | 82.9 | 68.0 | 48.6 | 65.1 |
| | IDOL [32] | Swin-L | 64.3 | 87.5 | 71.0 | 55.6 | 69.1 | 56.1 | 80.8 | 63.5 | 45.0 | 60.1 |
| | CTVIS [37] | Swin-L | 65.6 | 87.7 | 72.2 | 56.5 | 70.4 | 61.2 | 84.0 | 68.8 | 48.0 | 65.8 |
| | **SyncVIS** | Swin-L | **67.1** | **88.9** | **73.0** | **57.5** | **71.2** | **62.4** | **84.5** | **69.6** | **49.1** | **66.5** |
| Offline | EfficientVIS [30] | ResNet-50 | 37.9 | 59.7 | 43.0 | 40.3 | 46.6 | 34.0 | 57.5 | 37.3 | 33.8 | 42.5 |
| | IFC [17] | ResNet-50 | 41.2 | 65.1 | 44.6 | 42.3 | 49.6 | 35.2 | 55.9 | 37.7 | 32.6 | 42.9 |
| | Mask2Former-VIS [6] | ResNet-50 | 46.4 | 68.0 | 50.0 | - | - | 40.6 | 60.9 | 41.8 | - | - |
| | TeViT [36] | MsgShifT | 46.6 | 71.3 | 51.6 | 44.9 | 54.3 | 37.9 | 61.2 | 42.1 | 35.1 | 44.6 |
| | SeqFormer [31] | ResNet-50 | 47.4 | 69.8 | 51.8 | 45.5 | 54.8 | 40.5 | 62.4 | 43.7 | 36.1 | 48.1 |
| | VITA [15] | ResNet-50 | 49.8 | 72.6 | 54.5 | 49.4 | 61.0 | 45.7 | 67.4 | 49.5 | **40.9** | 53.6 |
| | DVIS [38] | ResNet-50 | 52.6 | 74.5 | **58.2** | 47.4 | 60.4 | 47.4 | 71.0 | 51.6 | 39.9 | 55.2 |
| | **SyncVIS** | ResNet-50 | **54.2** | **75.1** | **58.2** | **51.2** | **61.7** | **48.9** | **71.4** | **52.8** | 40.4 | **57.9** |
| | SeqFormer [31] | Swin-L | 59.3 | 82.1 | 66.4 | 51.7 | 64.4 | 51.8 | 74.6 | 58.2 | 42.8 | 58.1 |
| | Mask2Former-VIS [6] | Swin-L | 60.4 | 84.4 | 67.0 | - | - | 52.6 | 76.4 | 57.2 | - | - |
| | VITA [15] | Swin-L | 63.0 | 86.9 | 67.9 | 56.3 | 68.1 | 57.5 | 80.6 | 61.0 | 47.7 | 62.6 |
| | DVIS [38] | Swin-L | 64.9 | 87.0 | **72.7** | 56.5 | 69.3 | 60.1 | **82.0** | 67.4 | 47.7 | **65.7** |
| | **SyncVIS** | Swin-L | **65.7** | **87.3** | 72.5 | **56.7** | **69.8** | **60.3** | 81.8 | **67.5** | **48.6** | 65.4 |

single frame produces the frame embedding, there is no explicit video-level embedding to interact with the frame-level instance embedding. In our design, we add a set of video-level embeddings that gradually update with the frame-level embeddings. Our SyncVIS sets new state-of-the-art results under all of the settings. Among the online approaches, SyncVIS gets the highest performance of 57.9% AP and 67.1% AP with ResNet-50 and Swin-L backbones, which outperforms the previous best solution CTVIS [37] by 2.8 and 1.5 points, exceeds the top-ranking method DVIS [38] by 6.7 and 3.2 points, respectively. We list the model parameters and FPS of SeqFormer (220M/27.7), VITA (229M/22.8), and our SyncVIS (245M/22.1). Our model performs notably better with similar model parameters and inference speed. The two designs in SyncVIS can also boost the performance of both offline and online VIS solutions and can set new records in both settings, demonstrating the effectiveness and importance of synchronous modeling.

**YouTube-VIS 2021 & 2022.** Table 1 also compares the results on YouTube-VIS 2021. Our method hits the new records on the two backbone settings. SyncVIS achieves 51.9% AP and 62.4% AP with ResNet-50 and Swin-L backbones, respectively, outperforming the previous SOTA by 1.8 and 1.2 points, which further demonstrates the effectiveness of our approach. In Table 2, SyncVIS exceeds the previous SOTA by 1.1 points, proving its potency in handling complex long video scenarios.

**OVIS.** Table 3 illustrates the competitiveness of SyncVIS on the challenging OVIS dataset. SyncVIS also shows superior performance over other high-performance algorithms with 36.3% AP and 50.8% AP on ResNet-50 and Swin-L backbones, outperforming the current strongest architecture DVIS [38] by 2.2 and 0.9 points, respectively. SyncVIS harvests the highest performance on all four datasets, further evidencing its effectiveness and generality.

## 4.2 Ablation Studies

We ablate our proposed components, which are conducted with ResNet-50 on YouTube-VIS 2019.

Table 2. Results comparison on the YouTube-VIS 2022 long videos.

| | Method | AP | AP$_{50}$ | AP$_{75}$ | AR$_1$ | AR$_{10}$ |
|---|---|---|---|---|---|---|
| Swin-L | MinVIS [16] | 33.1 | 54.8 | 33.7 | 29.5 | 36.6 |
| | VITA [15] | 41.1 | 63.0 | 44.0 | **39.3** | 44.3 |
| | DVIS [38] | 45.9 | 69.0 | **48.8** | 37.2 | 51.8 |
| | SyncVIS | **47.0** | **69.4** | 48.6 | 38.9 | **52.4** |

Table 3. Results comparison on the OVIS.

| | Method | AP | AP$_{50}$ | AP$_{75}$ | AR$_1$ | AR$_{10}$ |
|---|---|---|---|---|---|---|
| R-50 | DVIS [38] | 34.1 | 59.8 | 32.3 | 15.9 | 41.1 |
| | SyncVIS | **36.3** | **60.9** | **33.0** | **17.0** | **42.8** |
| Swin-L | CTVIS [37] | 46.9 | 71.5 | 47.5 | 19.1 | 52.1 |
| | DVIS [38] | 49.9 | **75.9** | 53.0 | 19.4 | 55.3 |
| | SyncVIS | **50.8** | 75.7 | **53.1** | **20.5** | **55.9** |

Table 4. Experiments on aggregating our design to various popular VIS methods.

| Method | AP | Method | AP |
|---|---|---|---|
| Mask2Former-VIS [6] | 45.1 | VITA [15] | 49.5 |
| + Synchronized Modeling | 50.3 | + Synchronized Modeling | 53.0 |
| + Synchronized Optimization | 46.7 | + Synchronized Optimization | 51.2 |
| + Both (SyncVIS) | 51.5 | + Both (SyncVIS) | 54.2 |
| TMT-VIS [40] | 47.3 | DVIS [38] | 52.6 |
| + Synchronized Modeling | 51.1 | + Synchronized Modeling | 54.9 |
| + Synchronized Optimization | 48.7 | + Synchronized Optimization | 54.0 |
| + Both (SyncVIS) | 51.9 | + Both (SyncVIS) | 55.8 |
| GenVIS [14] | 51.3 | IDOL [32] | 49.5 |
| + Synchronized Modeling | 54.4 | + Synchronized Modeling | 55.1 |
| + Synchronized Optimization | 52.7 | + Synchronized Optimization | 51.3 |
| + Both (SyncVIS) | 55.4 | + Both (SyncVIS) | 56.5 |

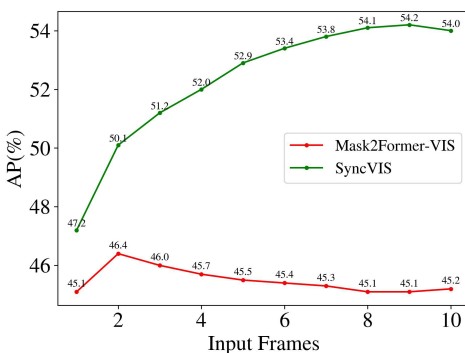

Fig. 3. Ablation study on the complexity of video scenarios regarding the number of input frames $T$.

Table 5. Ablation study on synchronized video-frame modeling.

| ID | Frame→Video | Video→Frame | AP | AP$_{50}$ | AP$_{75}$ |
|---|---|---|---|---|---|
| I | | | 45.1 | 65.7 | 49.0 |
| II | ✓ | | 50.2 | 72.5 | 54.2 |
| III | | ✓ | 48.6 | 71.4 | 51.8 |
| IV | ✓ | ✓ | 51.5 | 73.2 | 55.9 |

Table 6. Ablation study on the structure and query selection of synchronized video-frame modeling.

| Method | AP | AP$_{50}$ | AP$_{75}$ |
|---|---|---|---|
| Cascade Structure + Frame-level Queries | 46.2 | 67.8 | 49.9 |
| Cascade Structure + Video-level Queries | 46.7 | 68.2 | 50.3 |
| Cascade Structure + Both Queries | 49.9 | 72.0 | 54.4 |
| Synchronous Structure + Both Queries (SyncVIS) | 51.5 | 73.2 | 55.9 |

**Complexity of video scenarios.** Changing the complexity of video scenarios can check the capability of VIS solutions. We define the complexity as an indicator of the movements of different instances, which is calculated as the maximum combination of trajectories between frames. For example, if frame $t$ has $n$ instances while $t + 1$ frame has $m$, then the maximum complexity would be $mn$, and thus complexity is in polynomials with input frames, and we could use the frame number as an indicator of complexity. We examine the effect of different numbers of input frames in Fig. 3. We find the popular Mask2Former-VIS framework meets difficulties when dealing with complex videos, i.e., $T = 2$ works best for the model, and as $T$ continually increases, the performance will degrade notably. In contrast, as we increase the input frames, our SyncVIS improves gradually and achieves the best performance at $T = 9$. This evidences that our model is capable of handling challenging scenarios and can well characterize the movement trajectories of video instances.

**Key component designs.** Table 4 demonstrates the effect of our component designs when combined with the prevalent VIS methods. By aggregating the synchronized video-frame modeling paradigm, Mask2Former-VIS achieves a huge gain of 5.2 points in AP performance. This is credited to the design of two levels of queries as well as their mutual interactions. The synchronized embedding optimization strategy further advances performance improvement across all VIS methods. Aggregating two designs could also boost VITA by 3.5 and 1.7 points in performance, respectively. Note that the gain of 7.0 points for IDOL is also contributed by changing its original backbone to a Mask2Former-based backbone. The extensive results in Table 4 show that our new designs can introduce consistent improvements to various popular VIS methods, further indicating the effectiveness and generality.

## 4.3 Synchronized Video-Frame Modeling

**Enhancement direction.** In Table 5, we investigate the effect of the direction of the modeling paradigm, including synchronous bidirectional and asynchronous unidirectional ones. Unidirectional embedding enhancement can be divided into two types according to the output of the transformer decoder: i) utilize the frame-level embeddings to enhance the video-level ones, and the aggregation module consists of an FFN and cross-attention layer (denoted as 'Frame→Video'); ii) adopt the video-

level embeddings to update the frame-level ones, and feed the frame-level embeddings to prediction heads to generate the masks and instance classes independently (denoted as 'Video→Frame').

In Table 5, we find that without embedding enhancement, the decrease in performance is conspicuous as up to 6.4 points. With either unidirectional asynchronous embedding enhancement strategy, the result gets improved but is still not paired with the bidirectional synchronized video-frame modeling. This signifies several points: first, introducing frame-level embeddings to refine video-level embeddings can increment the performance by adding more frame-level instance details, thus strengthening the representative ability of video-level embeddings. Second, video-level embeddings contain more spatial-temporal information, and utilizing video-level embeddings to predict segmentation results for the video can receive better results. Third, adopting synchronized video-frame modeling is better than unidirectional modeling. Even though adding frame-specific information to video-level embeddings can contribute to representing more instance details, building the mutual association and aggregation leads to a stronger representation ability to characterize the semantics and motions.

**Modeling structure.** We suppose the superiority of using a synchronous structure over a cascade one is that the former avoids motion information loss and error accumulation. In Table 6, we evaluate these two structures. For the cascade structure, we use frame-level embeddings to extract information and associate image-level embeddings with video-level ones. The synchronized video-frame modeling and synchronized embedding optimization remain the same in cascade structure experiments. The synchronous structure gets 1.6 points higher AP performance than the cascade one, demonstrating the superior design of the proposed synchronous structure over the classical cascade structure.

**Query selection.** As shown in Table 6, utilizing only video-level queries performs better than only adopting frame-level ones. Frame-level queries segment each frame independently and focus less on the association across frames, which leads to lower performance. Our synchronous model, on the other hand, adopts both queries and achieves the best performance, validating the effectiveness of our synchronized video-frame modeling paradigm.

**Aggregation strategy.** Table 7 shows the results of different aggregation strategies in the synchronized video-frame modeling. In the 'Query Similarity', we select the most similar embeddings by computing the cosine similarity between video-level and frame-level embeddings. Note we compute similarities frame-by-frame and concatenate

Table 7. Ablation study on aggregation strategies.

| Method | AP | $AP_{50}$ | $AP_{75}$ | $AR_1$ | $AR_{10}$ |
|---|---|---|---|---|---|
| Query Similarity | 49.7 | 72.8 | 53.2 | 48.7 | 60.3 |
| Mask Similarity | 48.2 | 71.6 | 52.8 | 47.8 | 59.1 |
| Class Prediction | 51.5 | 73.2 | 55.9 | 49.5 | 60.4 |

the top $N_k$ embeddings together as input to the aggregation module. In the 'Mask Similarity', we get similarities of corresponding mask embeddings to determine the most similar ones. We use class scores (i.e., 'Class Prediction') to select key embeddings that work the best. Since some objects only appear in a few frames, the most similar embeddings may represent the background in extreme cases, disturbing the useful information for discrimination. Both aggregation methods have such problems, and using mask similarity is even worse since masks are insufficient to encode motion fully, leading to ineffective similarity calculation.

**Aggregation embedding size.** Table 8 shows the performance of SyncVIS with varying numbers of embedding in the aggregation stage of the synchronized video-frame modeling paradigm. When selecting top $N_k = 10$ embeddings to aggregate, the model performance reaches its best. When $N_k$ decreases, the aggregated key information contained in embeddings is not sufficient, the selected one may not encode the semantic information of all instances in the video, and therefore cause the drop in performance. Alternately, when $N_k$ gets larger than optimum, the redundant query features dilute the original information, which also leads to performance degradation.

### 4.4 Synchronized Embedding Optimization

**Sub-clips size.** Table 8 shows the results of SyncVIS with a varying $T_s$ of sub-clips. The larger the sizes of sub-clips are, the more complicated the optimization will be, and embeddings are less likely to capture the proper semantics and trajectories. When we set the size of sub-clips to 3, the model achieves its best performance. When $T_s$ decreases to the lowest, the problem of optimizing the whole video descends to optimizing each frame, weakening the model's ability to associate

Table 8. Ablation study on the aggregation embedding size $N_k$ of synchronized video-frame modeling paradigm and the sub-clip size $T_s$ of synchronized embedding optimization strategy.

| $N_k$ | AP | $AP_{50}$ | $AP_{75}$ | $AR_1$ | $AR_{10}$ | $T_s$ | AP | $AP_{50}$ | $AP_{75}$ | $AR_1$ | $AR_{10}$ |
|---|---|---|---|---|---|---|---|---|---|---|---|
| 5 | 51.1 | 73.0 | 55.4 | 49.1 | 59.3 | 1 | 50.9 | 73.7 | 54.9 | 49.0 | 60.1 |
| 10 | 51.5 | 73.2 | 55.9 | 49.5 | 60.4 | 2 | 51.3 | 73.3 | 55.6 | 49.2 | 60.2 |
| 25 | 50.9 | 73.5 | 55.1 | 48.4 | 59.6 | 3 | 51.5 | 73.2 | 55.9 | 49.5 | 60.4 |
| 50 | 49.3 | 72.8 | 52.3 | 47.4 | 56.7 | 4 | 50.7 | 73.8 | 54.1 | 47.9 | 58.9 |
| 100 | 47.5 | 70.4 | 51.4 | 46.8 | 56.1 | 5 | 50.4 | 73.3 | 54.2 | 47.2 | 58.1 |

Table 9. Ablation study on synchronized embedding optimization strategy with ResNet-50 backbone.

| Datasets | Method | AP | $AP_{50}$ | $AP_{75}$ |
|---|---|---|---|---|
| YouTube-VIS 2019 | Mask2Former-VIS | 45.1 | 65.7 | 49.0 |
| | + Optimization | 46.7 | 68.6 | 50.7 |
| YouTube-VIS 2021 | Mask2Former-VIS | 39.8 | 59.8 | 41.5 |
| | + Optimization | 41.3 | 62.1 | 42.5 |
| OVIS | Mask2Former-VIS | 10.6 | 25.4 | 7.2 |
| | + Optimization | 12.3 | 27.1 | 9.2 |

frames temporally. When $T_s$ increases, though there is a gain in the performance when compared to undivided circumstances, the optimization is still more complex, making the training process hard to reach optimum. Learned embeddings are insufficient to capture all semantics for sub-clip, and therefore the performance is weaker than the optimal $T_s$ value. However, $T_s = 3$ is the optimum for Youtube-VIS 2019 & 2021. For Youtube-VIS 2022 and OVIS, SyncVIS performs best when $T_s$ is 2, which is the smallest size to maintain temporal associations. We suppose, that for more complex scenarios, dividing into smaller sub-clips is beneficial for query embeddings to associate across frames and accelerate the optimization. In optimization strategy, our main goal is to reduce the increasing optimization complexity as the input frame number grows. To realize this target, our strategy is to divide the video into several sub-clips that could make optimization easier while retaining the temporal motion information. Longer Sub-clips could provide the model with more temporal information, but their optimization complexity also rises polynomially. By optimizing sub-clips, models can better adapt to changes in the target instance within the video, particularly in cases of occlusion of many similar instances (In OVIS, most cases are videos with many similar instances, most of which are occluded in certain frames). By optimizing the local sub-sequence of the video, rather than the entire video sequence, if the target instance becomes occluded in certain frames, our optimizing strategy can adjust the features within the sub-sequence to adapt to this change, without being affected by the unoccluded frames.

**Generality.** The proposed optimization strategy is effective and general that can be adapted into various DETR-based approaches. In these frameworks, the optimization problem for long video sequences still exists. As in Table 9, when adding our optimization strategy to Mask2Former-VIS, we harvest notable performance gains on all three benchmarks. This demonstrates that the proposed optimization can be treated as a robust design suitable for different video scenarios.

## 5 Conclusion

We have proposed SyncVIS for synchronized Video Instance Segmentation. Unlike the current VIS approaches that use asynchronous structures, SyncVIS utilizes a synchronized video-frame modeling paradigm to encourage the synchronization between frame embeddings and video embeddings in a synchronous manner, which incorporate both semantics and movement of instances more effectively. Moreover, SyncVIS develops a plug-and-use synchronized embedding optimization strategy during training, which reduces the complexity of bipartite matching in a divide-and-conquer approach. Based on these two designs, our SyncVIS outperforms current methods and achieves SOTA on four challenging benchmarks. We hope that our method can provide valuable insights and motivate the future VIS research.

**Broader impacts and limitations.** SyncVIS is designed to propose a new synchronized structure for VIS with promising performance. We hope this work can contribute to further applications in video-related tasks and real-life applications. However, even though our model achieves promising results, it has a problem segmenting very crowded or heavily occluded scenarios, which is discussed in the supplementary.

**Acknowledgement.** This work is partially supported by the National Natural Science Foundation of China (No. 62201484), National Key R&D Program of China (No. 2022ZD0160100), HKU Startup Fund, and HKU Seed Fund for Basic Research.

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

## Appendix

This appendix provides more details about the proposed SyncVIS, more qualitative visual comparisons, and the codebase of our implementation. The content is organized as follows:

- More ablation study experiments of the SyncVIS.
- The qualitative visual comparisons between popular VIS methods and our SyncVIS.
- The codebase is contained in the link:
  https://github.com/rkzheng99/SyncVIS

## A  Dataset Details

Here, we provide a detailed overview of various VIS datasets in Table 10. Our extensive experimental evaluations are conducted on four challenging benchmarks, namely YouTube-VIS 2019, 2021, and 2022 [34], and OVIS [27]. YouTube-VIS 2019 [34] was the first large-scale dataset designed for video instance segmentation, comprising 2.9K videos averaging 4.61s in duration and 27.4 frames in validation videos. YouTube-VIS 2021 [34] poses a greater challenge with longer and more complex trajectory videos, averaging 39.7 frames in validation videos. The OVIS [27] dataset is another challenging VIS dataset with 25 object categories, focusing on complex scenes with significant object occlusions. Despite containing only 607 training videos, OVIS's videos last an average of 12.77s. Lastly, the most recent update, YouTube-VIS 2022, adds an additional 71 long videos to the validation set and 89 extra long videos to the test set.

## B  Additional Ablation Studies

**Update momentum.**  In this part, we show the performance of different values of $\lambda$, which is the update momentum in the synchronized video-frame modeling module. When $\lambda$ equals zero, the whole synchronization between two levels of embeddings is collapsed, and thus a huge degradation in performance is shown in Table 11. As the $\lambda$ grows larger than the optimum value, the synchronization can not bring further gain. Rather, the aggregation interferes with the updating of both levels of embeddings in the decoder, which leads to a less increase in performance. Noted that in this experiment, we base our approach on **IDOL** instead of Mask2Former or CTVIS.

**Limitations.**  As for limitations, our model has a problem in segmenting very crowded or heavily occluded scenarios. Even though our model shows better performance in segmenting complex scenes with multiple instances and occlusions than previous approaches (as shown in visualizations in the main paper and supplementary file), handling with extremely crowded scenes is not our main focus. Our SyncVIS, on the other hand, aims to build consistent video modeling by synchronously implementing both video-level and frame-level embeddings as well as synchronized optimizations. We provide visualizations in our github repo: https://github.com/rkzheng99/SyncVIS.

## C  Visualization

Visual comparisons of different VIS methods are illustrated in Fig. 4. Our proposed SyncVIS obtains accurate segmentation masks and captures occluded movement trajectories in challenging video scenarios, evidencing its effectiveness over traditional solutions. In the visualization comparisons between Mask2Former-VIS and our model, we select some cases under different scenarios, which include setting with multiple similar instances, setting with reappearance of instance, setting with different poses of instance, and settings with long video in Fig. 5, Fig. 6 and Fig. 7. The high-quality segmentation results under these diverse circumstances and scenarios prove our model's robustness and generality in modeling both semantics and movements of objects. Also, we choose visualizations of implementing different levels of embeddings in Fig. 8. The comparisons further prove the effectiveness of synchronized video-frame modeling.

Table 10. Key statistics of popular VIS datasets.'YTVIS' is the acronym of 'Youtube-VIS'.

| | YTVIS19 | YTVIS21 | OVIS |
|---|---|---|---|
| Videos | 2883 | 3859 | 901 |
| Categories | 40 | 40 | 25 |
| Instances | 4883 | 8171 | 5223 |
| Masks | 131K | 232K | 296K |
| Masks per Frame | 1.7 | 2.0 | 4.7 |
| Object per Video | 1.6 | 2.1 | 5.8 |

Table 11. Ablation study of $\lambda$ in synchronized video-frame modeling paradigm. The results are evaluated on the Youtube-VIS 2019 dataset.

| $\lambda$ | AP | $AP_{50}$ | $AP_{75}$ | $\lambda$ | AP | $AP_{50}$ | $AP_{75}$ |
|---|---|---|---|---|---|---|---|
| 0.0 | 52.5 | 74.8 | 57.3 | 0.10 | 56.1 | 78.8 | 59.3 |
| 0.01 | 54.9 | 77.6 | 58.7 | 0.12 | 55.7 | 78.3 | 58.9 |
| 0.02 | 55.8 | 78.9 | 59.0 | 0.15 | 55.2 | 78.1 | 58.4 |
| 0.05 | 56.5 | 79.5 | 59.8 | 0.20 | 54.3 | 77.4 | 58.0 |
| 0.08 | 56.3 | 79.1 | 59.2 | 0.50 | 53.0 | 75.1 | 57.8 |

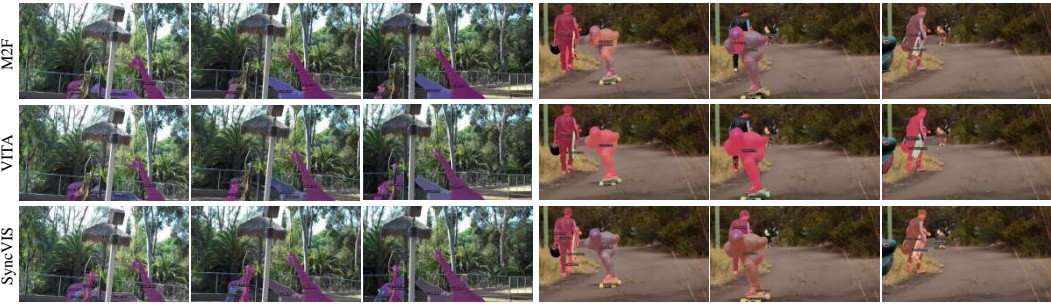

Fig. 4. Visual comparison of our SyncVIS with Mask2Former-VIS ('M2F') [6] and VITA [15]. SyncVIS shows impressive accuracy in long, complex scenarios where objects share similar appearances and have heavy occlusions.

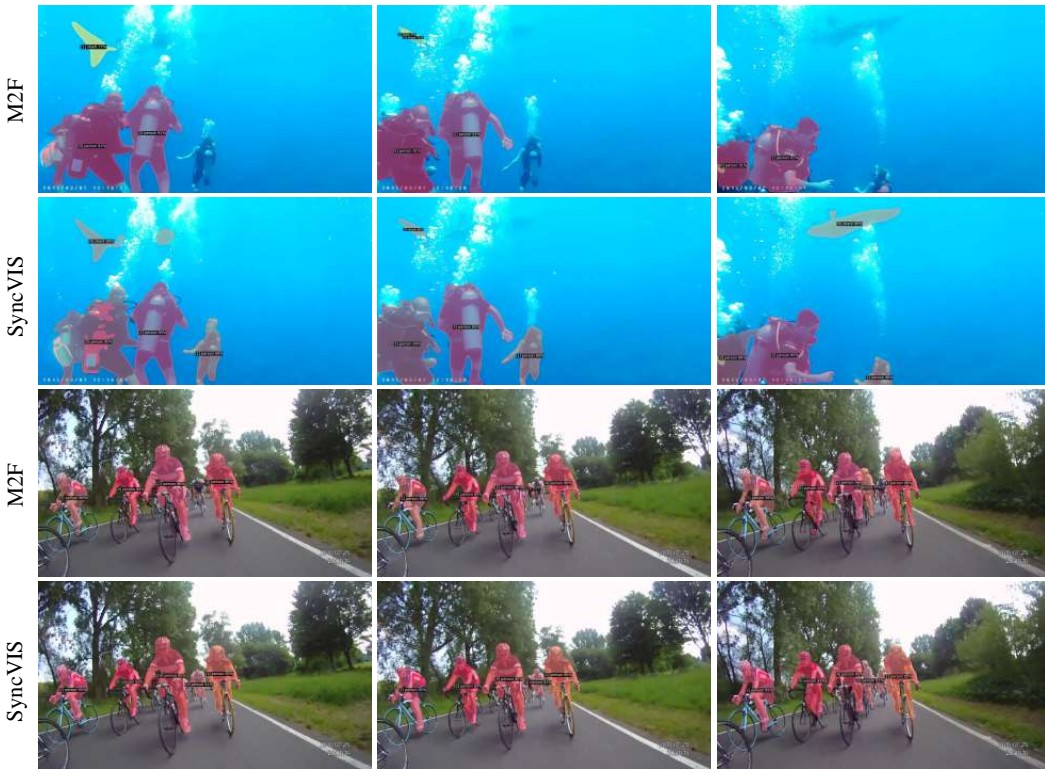

Fig. 5. Qualitative comparisons with Mask2Former-VIS (abbreviated as 'M2F') on Youtube-VIS 2019. In this case, we want to further prove that SyncVIS can better distinguish and capture instances with the same identities. In the first two rows, the person on the right is not segmented by Mask2Former-VIS, while in the last two rows, the cyclist from the back is not segmented by Mask2Former-VIS.

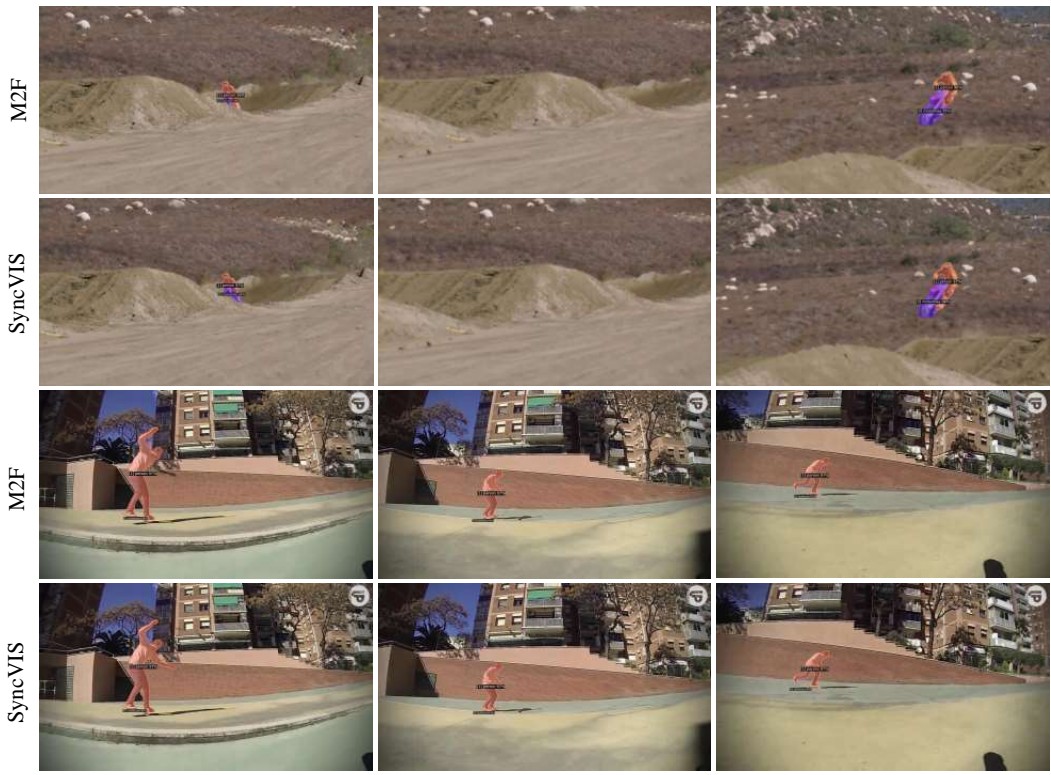

Fig. 6. Qualitative comparisons with Mask2Former-VIS (abbreviated as 'M2F') on Youtube-VIS 2019. In the first two rows, the person riding the motorcycle reappears in the frame, which tests the model's ability to connect instances across the temporal axis. Our model successfully connects instances across two frames and segments more precisely than Mask2Former-VIS does (the motorcyclist's leg in the third frame), which demonstrates SyncVIS's temporal association ability. In the last two rows, the person on the skateboard is changing his poses across time. Our model successfully segments the person in different poses, while Mask2Former-VIS fails to segment this person's arm in the first frame. This further illustrates the robustness and generality of our model's temporal association ability, which is credited to the synchronization.

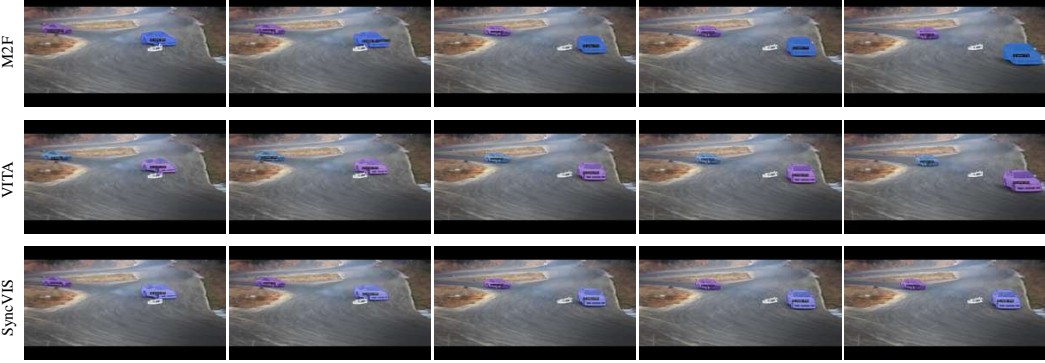

Fig. 7. Visual comparison of our SyncVIS with Mask2Former-VIS (abbreviated as 'M2F') [6] and VITA [15]. SyncVIS shows impressive accuracy in long videos, while the previous methods have either low confidence (the confidence of the car in blue masks in the first row is 77% while in the third row is 98%) or incomplete masks (the first frame in the second row).

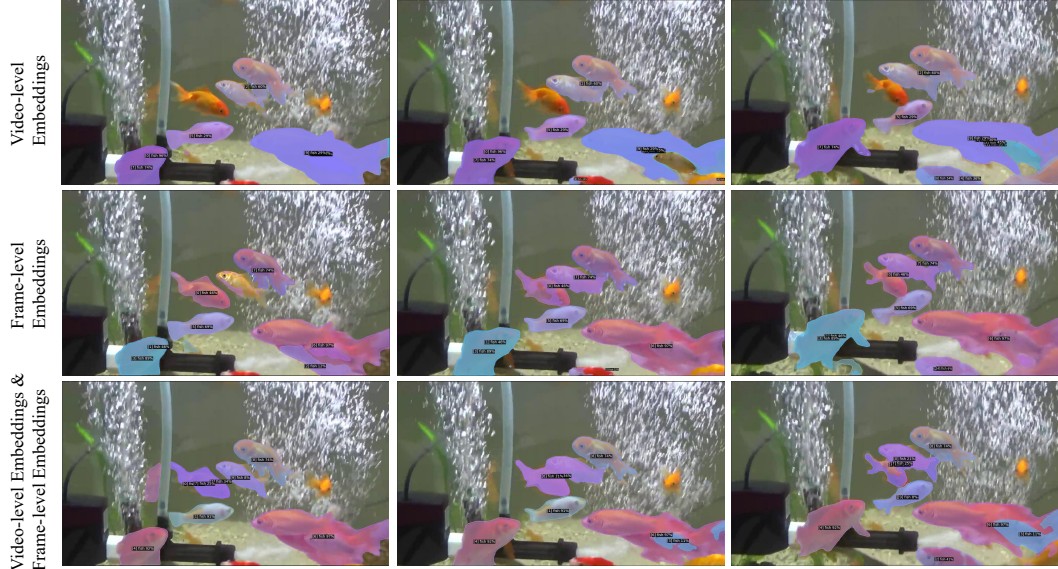

Fig. 8. Qualitative comparisons with different designs of embeddings. Video-level embeddings are from a set of shared instance queries for all sampled frames. In the first row, video-level embeddings successfully capture most of the instances, but fail to mask the fish in the middle of the image. Frame-level embeddings are assigned to each sampled frame. In the second row, frame-level embeddings segment instances better than the first row, but fail to maintain the trajectories of fish in the bottom right. When synchronizing these two sets of embeddings, our model achieves better segmentation results even under such a complex scenario.

