# OpenReview forum: "SyncVIS: Synchronized Video Instance Segmentation"
_NeurIPS.cc/2024/Conference — NeurIPS 2024 poster_

### Official Review · Reviewer_hMrY · 2024-07-07

**Soundness:** 2
**Presentation:** 2
**Contribution:** 2
**Rating:** 5
**Confidence:** 4

**Summary:**

The paper argues that existing methods for video instance segmentation use asynchronous designs, leading to difficulties in handling complex video scenarios. To address this problem, the paper proposes a SyncVIS method for synchronized modeling, achieving state-of-the-art results on several benchmarks.

**Strengths:**

1. The paper is generally clear to understand.
2. The paper achieves good performance by synchronized modeling.
3. Visualizations are provided to illustrate the effectiveness of the method.

**Weaknesses:**

1. The novelty of synchronized embedding optimization is limited.
2. The paper did not compare with DVIS++. For example, using ResNet-50 and offline mode, DVIS++ outperforms SyncVIS by 2.5% on YouTube-VIS 2019. DVIS++: Improved Decoupled Framework for Universal Video Segmentation, arXiv, 2023.
3. More implementation details like training steps should be reported.
4. Compute resources, such as the type of GPU, are not reported in the paper, yet the authors answered YES to question 8 in the checklist.

**Questions:**

In addition to the Weaknesses,
1. How does the method ensure that the video queries learn motion information?
2. Ablation study on different values of $N_{k}$ (line 173) should be conducted.
3. How is the efficiency of the method?

**Limitations:**

Yes.

---

> ### Author Rebuttal · Authors · 2024-08-07
>
> In the beginning, we want to thank you for the detailed, insightful, and constructive comments.
>
> ### Novelty
>
> To sum up, as other reviewers have mentioned, our method provides an architectural design that is **intuitive and effective** (Reviewer 3JCg) and **innovative** (Reviewer 2sLS), **demonstrating a high level of versatility** (Reviewer MsYp). Our SyncVIS explicitly introduces video-level query embeddings to synchronize video-level embeddings with frame-level query embeddings. The synchronized video-frame modeling paradigm promotes the mutual learning of frame- and video-level embeddings by selecting key information from both embeddings and updating gradually via the synchronous decoder structure.
>
> ### DVIS++
>
> In offline mode, our method is lower than DVIS++, but our **best** performance on ResNet-50 backbone is in **online** mode, which is higher than DVIS++, by adapting to CTVIS. As mentioned in our paper and credited by Reviewer MsYp, our method **demonstrates a high level of versatility** on many state-of-the-art VIS methods. DVIS++ is built on DVIS, which incorporates a denoising training strategy and contrastive learning paradigm as well as an improved DINOv2 backbone. These adjustments, however, are orthogonal to our improvements to the modeling of spatial-temporal representations as well as our optimization strategy.
>
> In the referring tracker and the temporal refiner, with the addition of our video-level embeddings and our synchronous modeling structure, our method can still bring significant improvements (2.1 AP) by adapting to DVIS++ in offline mode.
>
> | Method                      | AP   |
> |-----------------------------|------|
> | DVIS++                      | 56.7 |
> | + Synchronized Modeling     | 58.0 |
> | + Synchronized Optimization | 57.6 |
> | + Both (SyncVIS)            | 58.8 |
>
> ### Implementation details
> As for the implementation details such as training steps, we list these parameters in the **configs** file in the link provided in the **Appendix**. For example for the Youtube-VIS 2019, as the batch size is set to 8, the max_iter is 140000, and we use AdamW optimizer with a base learning rate of 5e-4 and a weight decay of 0.05. The backbone multiplier is 0.1.
>
> ### Computation resources
> Most of our experiments are conducted on 4 A100 GPUs (80G), and on a cuda 11.1, PyTorch 3.9 environment. The training time is approximately 1.5 days when training with the Swin-L backbone.
>
> ### Motion information in video queries
> In the DETR-style architecture, when video queries are **associated with features across time** via the decoder, they can effectively model instance-level motion through the cascade structure. In Mask2Former-VIS, the use of video queries alone enables the capture of instance motion.
>
> A similar finding has been reported in the Seqformer model, which notes that **"a stand-alone instance query suffices for capturing a time sequence of instances in a video."** However, as the number of input frames increases, relying solely on video queries becomes insufficient for simultaneously tracking the movement of all instances. Such limitations motivate us to propose such synchronous designs to address the shortcomings in motion modeling.
>
> ### Ablation of $N_k$
> The ablation study of N_k is provided in Table 8 and the related analysis starts in L.324. The ablation results reveal that when selecting top $N_{k}=10$ embeddings to aggregate, the model performance reaches its optimum. When $N_{k}$ gets larger than optimum, the **redundant** query features will **dilute** the original compact information. On the other hand, getting too small will lead to the insufficiency of the injected information.
>
> ### Efficiency
> The relevant results are provided in L.242-243. We list the model parameters and FPS of SeqFormer (220M/27.7), VITA (229M/22.8), and our SyncVIS (245M/22.1). Our model performs **notably better** with comparable model parameters and inference speed.

---

> > ### Author Response · Authors · 2024-08-11
> >
> > Dear Reviewer hMrY,
> >
> > We appreciate the time and effort you have dedicated to reviewing our work. Your feedback is invaluable, and we are grateful for the opportunity to address any concerns you may have. If you have any further questions or require additional clarification on our rebuttal, please do not hesitate to reach out to us. We stand ready to provide any necessary information or explanations to facilitate your review process.
> >
> > Thank you again for your thorough review. We look forward to receiving your feedback and are hopeful for a favorable consideration of our work.

---

> > > ### Comment · Reviewer_hMrY · 2024-08-14
> > >
> > > Thank you for the authors' rebuttal. Most of my concerns have been addressed. I hope the authors will include all the experiments raised by the reviewers in the final version and present important details, like FPS, more explicitly in the table rather than in a short sentence.

---

### Official Review · Reviewer_2sLS · 2024-07-09

**Soundness:** 2
**Presentation:** 3
**Contribution:** 3
**Rating:** 5
**Confidence:** 5

**Summary:**

This paper focuses on improving synchronization between frame and video queries in video instance segmentation for better long-range video analysis. The authors propose encoding frame and video queries separately, then using confidence scores to select Nk queries. These queries are updated through mutual information exchange with momentum updates. To train each query, they introduce a synchronized optimization method using a divide-and-conquer approach. They also identify that video-level bipartite matching complexity increases with the number of frames, and address this by suggesting sub-clip level matching. Their technique, when applied to CTVIS and VITA, demonstrates enhanced performance in both online and offline settings compared to existing methods.

**Strengths:**

- The paper is comprehensively written, with clear explanations of its contributions and a detailed analysis of prior research. The proposed methods are innovative and seem well-founded.

- Extensive experiments validate the effectiveness of the proposed methods, significantly bolstering the paper's credibility.

**Weaknesses:**

- The analysis primarily focuses on early work (Mask2Former-VIS), while the baselines used are CTIVS and VITA. Despite this, there is a lack of thorough analysis on these baselines.
- The GT assignment method of Synchronized Embedding Optimization is not compared with existing methods such as TCOVIS. Additionally, although the authors aim for long-range video modeling, their performance on the long video dataset YouTube-VIS 2022 is lower than that of TCOVIS.
- In the Checklist under Experiments Compute Resources, the authors answered "yes" but did not specify the equipment used.


Reference

Li, Junlong, et al. "Tcovis: Temporally consistent online video instance segmentation." Proceedings of the IEEE/CVF International Conference on Computer Vision. 2023.

**Questions:**

- In Section 3.1, the author highlights the performance drop in Mask2Former-VIS when increasing the number of frames. CTVIS, on the other hand, shows improved performance with more frames and seems to handle temporal modeling well through its memory bank. If CTVIS were used as the baseline, why does the proposed method show improved performance and what specific aspects are enhanced?

- TCOVIS aggregates the cost for each frame and matches GT with predictions at the video level globally. The proposed method matches at the sub-clip level, which is a contrasting approach. What are the respective strengths and weaknesses of each method? Additionally, why does the proposed method perform worse on long videos like YouTubeVIS-2022 compared to TCOVIS?

- In L219, it is mentioned that inference with the Swin-L backbone was done at 480p, but the code indicates 448p. Which is correct? Furthermore, in L218, the learning rate is stated as 1e-4, but the code shows 5e-5. Also, the iteration and learning rate decay settings for YouTubeVIS-2019 and 2021 datasets seem inconsistent. How were the optimization settings determined?

**Limitations:**

The authors address the limitations of the proposed method in the appendix.

---

> ### Author Rebuttal · Authors · 2024-08-07
>
> We appreciate the reviewer's thorough review and valuable input. Your feedback has been instrumental in helping us enhance the quality and clarity of our work.
>
> ### Analysis of baseline method
>
> The analysis of CTVIS and VITA are included in the Introduction and Related Works (L. 37, L. 104). We choose to analyze Mask2Former-VIS because it's a basic and representative VIS model that both VITA and CTVIS are built on, while VITA utilizes frame-level queries to segment each frame independently and then associate frame-level queries with video-level queries and CTVIS adopts the frame-level queries to build consistent contrastive items and memory banks. 1) **VITA**: Even though VITA adopts video-level queries and manages to associate them, we argue that this process is **sensitive** because the wellness of video-level queries **heavily relies on the learning of frame-level queries**. Since frame-level queries (decoded by frame features) lack global motion information, this asynchronous unidirectional aggregation will cause potential information loss to the output video-level queries. 2) **CTVIS**: CTVIS utilizes contrastive learning to strengthen the representations of frame-level embeddings and uses frame-level embeddings to maintain a memory bank to model temporal relations. However, it mainly focuses on building discriminative frame-level embeddings, and **hardly models the long-term spatio-temporal object representation**. That is, CTVIS **lacks an explicit synchronous association** between video- embeddings and frame-level embeddings. As proved in YoutubeVIS 2019 & 2021, SyncVIS achieves 57.9 and 51.9 in the online mode, while CTVIS is 55.1 and 50.1, which is lower than our results.
>
> We will revise our paper in the final version.
>
>
> ### Comparison with TCOVIS
>
> In TCOVIS, the global instance assignment aims to match predictions from all frames in the video clip with GTs. It considers all frames as a whole in searching for the optimal objective. Our method, on the other hand, divides all video frames into smaller sub-clips for matching with GTs.
>
> 1) **TCOVIS:** By matching predictions across all frames, TCOVIS is capable of **associating the video frames from the beginning to the end** of a minute-long video in the Youtube-VIS 2022 validation dataset, but it may **overlook the fine-grained local details** that is crucial for distinguishing multiple similar instances. Thus, when handling multiple instances and movements in consecutive frames, TCOVIS fails to simultaneously track and segment many more instances in crowded and occluded scenes. On OVIS, TCOVIS is 46.7 while our SyncVIS is 50.8, outperforming by a large margin.
> 2) **SyncVIS:** By dividing video into several sub-clips, our optimization strategy aims to **reduce the increasing optimization complexity** as the input frame number grows (L. 188, L. 260) because our synchronous modeling paradigm can model temporal association better with more input frames. To realize this target, our strategy is to divide the video into several sub-clips that could make optimization easier while retaining the temporal motion information.
>
> 3) **YoutubeVIS 2022:** Our SyncVIS mainly focuses on solving challenging scenarios with multiple instances and occlusions, which is usually overlooked for previous asynchronous methods because of their query-sensitive designs. When handling minute-long videos, our video-level embeddings are insufficient for modeling the associations across such a huge amount of input frames. Therefore, the performance on Youtube-VIS 2022 is not the SOTA result.
>
>     However, our SyncVIS manages to find a balance between modeling long-range video and keeping high performance on crowded and occluded video scenes in shorter lengths. Even though TCOVIS has better results on Youtube-VIS 2022 validation dataset, its performance on another challenging VIS dataset, OVIS, is 4.1 points below our SyncVIS. TCOVIS manages to model the extra-long videos in Youtube-VIS 2022 validation dataset, but it somehow neglects the much more common cases, which are combinations of long videos and complex scenes with more instances and occlusions.
>
> We will revise this part to the final version of the paper.
>
>
> ### Computation resources
>
> Most of our experiments are conducted on 4 A100 GPUs (80G), and on a cuda 11.1, PyTorch 3.9 environment. The training time is approximately 1.5 days when training with the Swin-L backbone. We will specify this in the final version of the paper.
>
> ### Optimization setting
>
> Since the two datasets have different sizes, and that 2021 has more videos than 2019, we use different training iterations and learning rates for different datasets. in practice.  We will revise our paper in the final version to state this difference (As for the image size, the number should be 448, and the learning rate of the Swin-Large backbone on YoutubeVIS 2019 is 5e-4, and we will correct this in the final version).
>
> ### Improvement upon CTVIS
>
> It's true that building a memory bank contributes to the temporal modeling of instances. However, the forward passing of CTVIS is still asynchronous. While a single frame produces the frame embedding, there is **no explicit video-level embedding** to interact with the frame-level instance embedding. In our design, we add a set of video-level embeddings that gradually update with the frame-level embeddings. The explicit video-level embedding can directly provide long-range information and temporal information to the frame-level embedding, which enhances the quality of Hungarian matching and the contrastive learning module afterward.

---

> > ### Comment · Reviewer_2sLS · 2024-08-09
> >
> > Thank you for the authors' thorough responses. Now, I understand and agree that bidirectional communication between video queries and frame queries is crucial for solving the VIS task, and that this is a point overlooked by previous works.
> >
> > However, I am still not convinced about the sub-clip matching approach.
> >
> > - I don't understand why matching predictions and ground truth at the sub-clip level would lead to better distinction between multiple instances. Intuitively, it seems that globally matching predictions to each object's trajectory would be more effective.
> > - Since TCOVIS is based on GenVIS and SyncVIS is based on CTVIS, I find it difficult to compare their performance on OVIS alone as a measure of optimization effectiveness. I'm not confident that sub-clip matching optimizes better than global matching for the OVIS dataset.
> > - Also, the memory issue doesn't seem significant since VIS tasks typically don't use a large number of frames for ground truth assignment during the training phase.

---

> > > ### Author Response · Authors · 2024-08-10
> > >
> > > Thank you for taking the time to review our rebuttal and sharing your perspectives. We are glad that the significance of our synchronous design is acknowledged, and we'd love to address to your additional comments.
> > >
> > > ### Advantage of sub-clip level matching
> > >
> > > - The advantage of optimizing sub-clip over the GIA strategy (TCOVIS) is its ability to **better adapt to changes** in the target instance within the video, particularly in cases of occlusion of many similar instances (In OVIS, most cases are videos with many similar instances, most of which are occluded in certain frames.)
> > >
> > > - By optimizing the local sub-sequence of the video, rather than the entire video sequence, if the target instance becomes occluded in certain frames, our optimizing strategy can adjust the features within the sub-sequence to adapt to this change, **without being affected by the unoccluded frames**. For instance, if the target becomes occluded between frames 10 and 15. For the GIA strategy, it would be difficult to adjust the features in these last 5 frames, as the features in the initial 10 frames may be well-represented.
> > >
> > > - We tested two optimization methods based on GenVIS in the online setting with ResNet-50 backbone on OVIS dataset. As shown in the table, our method shows **better improvement** (0.6 AP) over GIA. This result is consistent with our analysis that our optimization strategy is more effective under such scenarios.
> > >
> > >
> > > | Method                                                         | $AP$ | $AP_{50}$ | $AP_{75}$ |
> > > |----------------------------------------------------------------|:--:|:-----:|:-----:|
> > > | GenVIS                              |35.8| 60.8  |  36.2 |
> > > | GenVIS + Global Instance Assignment |36.2| 61.4  |  36.6 |
> > > | GenVIS + Synchronized Optimization  |36.8| 62.3  |  37.0 |
> > >
> > > ### Memory issue
> > > - As for the memory issue, previous VIS methods would take input frame number at a quite low value (For example, Mask2Former is 2). However, we argue that this is because their performance will even drop (due to their asynchronous structure) when handling more input frames during training (as shown in Fig. 3). But in our design, we introduce an efficient synchronous modeling paradigm that is capable of efficiently utilizing temporal information with more input frames. With more input frames, the memory issue becomes more significant.

---

> > > > ### Author Response · Authors · 2024-08-13
> > > >
> > > > Dear Reviewer 2sLS,
> > > >
> > > > We sincerely appreciate your dedicated time and effort in reviewing our work. Your valuable comments and feedback are crucial for improving the quality of our research.
> > > >
> > > > We have carefully considered your concerns and provided corresponding responses and updated results. We believe we have addressed the key issues you raised, and we welcome further discussion to ensure we have fully covered your concerns. Please let us know if any remaining unclear parts or areas require additional clarification. We are committed to providing a comprehensive response and are open to any additional feedback you may have. Your input is invaluable in helping us strengthen our work.
> > > >
> > > > Thank you again for your thorough review. We look forward to continuing our productive discussion and are hopeful for a favorable consideration of our work.

---

> > > > > ### Comment · Reviewer_2sLS · 2024-08-13
> > > > >
> > > > > Thank you for your response. Since most of my concerns have been addressed, I will raise the score to BA. In the main text, the discussion on optimization mainly focused on memory issues, making it difficult to identify which aspects contributed to performance improvement. I hope the final version will be more organized in this regard.

---

> > > > > > ### Author Response · Authors · 2024-08-13
> > > > > >
> > > > > > Dear Reviewer 2sLS,
> > > > > >
> > > > > > Thank you very much for your positive feedback! We are glad to hear that our rebuttal has resolved most of your concerns! As for the discussion on optimization in the context, we will revise this part in the final version. Again, we thank the reviewer for the valuable suggestions, which have been very helpful in improving our work.

---

### Official Review · Reviewer_MsYp · 2024-07-13

**Soundness:** 3
**Presentation:** 2
**Contribution:** 2
**Rating:** 5
**Confidence:** 5

**Summary:**

This paper concentrates on the video instance segmentation task. To address the problem of motion information loss when existing methods use frame queries, and the optimization issue of bipartite matching across multiple frames, the authors propose a synchronized video-frame modeling paradigm. This paradigm allows for interaction between frame queries and video queries and introduces a synchronized embedding optimization strategy in a video-level buffer, making multi-frame tracking optimization feasible. The effectiveness of the proposed method is verified on four VIS benchmarks, and detailed ablation studies are conducted.

**Strengths:**

1. The proposed SyncVIS shows impressive performance across multiple VIS benchmarks.
2. The method introduced can be adapted to enhance multiple existing VIS frameworks, demonstrating a high level of versatility.

**Weaknesses:**

1. Line 45 mentions that 'image encoding stage (rather than video encoding)' could lead to motion information loss; however, the proposed method still employs image encoding. Frame queries take video query for feature aggregation, the aggregation is merely an accumulation of multi-frame information, similar to the video query in SeqFormer, which can help achieve robustness, but how does it model object motion? This statement might be inappropriate.

**Questions:**

1. The proposed Synchronized Embedding Optimization for bipartite matching is it aimed at the bipartite matching between prediction and Ground Truth, or object query bipartite matchings during the tracking process (Like MinVIS)? This aspect is quite confusing for me. From lines L186-L193, it seems that the author aims to enhance bipartite matching between prediction and GTs. Yet, lines L194-L199 appear to discuss associating the tracking results of two frames.

2. The conclusion in Sec4.4 seems counterintuitive. Longer Sub-clips could provide the model with more temporal information to help it model complex scenes and trajectories. This should be especially evident on OVIS, which includes many objects and complex inter-object motions, including occlusions and disappearances and reappearances. According to the paper's motivation, longer Sub-clips should perform better on OVIS, but the experimental conclusion states the optimal length on OVIS to be T_s = 2, which corresponds to a pair of frames, providing very limited temporal information. Doesn't this contradict the motivation or the conclusion?

**Limitations:**

Yes

---

> ### Author Rebuttal · Authors · 2024-08-07
>
> We are grateful to the reviewer for taking the time to provide such detailed and constructive criticism. Your suggestions have been invaluable in strengthening our paper.
>
> ### Object motion modeling
>
> - **Synchronous design for robust modeling**
>
> Our proposed SyncVIS employs image encoding as well as video encoding **in a synchronous manner** to model object motion while compensating for the potential "motion information loss" in the "image encoding" stage. In our design, frame-level embeddings are assigned to each sampled frame, and responsible for modeling the appearance of instances, and video-level embeddings are a set of shared instance queries for all sampled frames, which are used to characterize the general motion (because they encode the position information of instances across frames, and thereby naturally contain the motion information).
>
> SeqFormer has the observation that "a stand-alone instance query suffices for capturing a time sequence of instances in a video." It decomposes the decoder to be frame-independent while building communication between different frames using instance queries, which also aim to model motion across frames. Nevertheless it follows an asynchronous manner, which is **less robust** in modeling temporal associations than our synchronous design because the aggregation is unidirectional instead of a mutual enhancement and the motion loss in image encoding is not compensated. As proved in YoutubeVIS 2019 & 2021, SyncVIS achieves 54.2 and 48.9 in the offline mode, while Seqformer is 47.4 and 40.5, which is much lower than our results.
>
> - **Experiment**
>
> As shown in Table (also in the Table 6 of the main paper), by implementing a synchornous structure, SyncVIS outperforms the asynchronous unidirectional structure by 1.6 AP. This further proves that our design is more robust than the design in SeqFormer in modeling object motion.
>
>
> | Method                                                         | $AP$ | $AP_{50}$ | $AP_{75}$ |
> |----------------------------------------------------------------|:--:|:-----:|:-----:|
> | Cascade Structure + Both Queries                               |49.9| 72.0  |  54.4 |
> | Synchronous Structure + Both Queries (SyncVIS)                 |51.5| 73.2  |  55.9 |
>
> ### Synchronized Embedding Optimization
>
> In the synchronized embedding optimization, we aim to enhance bipartite matching between prediction and GTs. L194-L199, the illustrations of $t_i$ and $t_j$ are trying to explain how the optimization strategy works in a divide-and-conquer way and illustrate our motivation of dividing the whole training sample videos into sub-clips.
>
> ### Setting of $T_s$
>
> The results do not contradict the conclusion. In optimization strategy, our main goal is to **reduce the increasing optimization complexity** as the input frame number grows (L. 188, L. 260). To realize this target, our strategy is to divide the video into several sub-clips that could make optimization easier while retaining the temporal motion information. Longer Sub-clips could provide the model with more temporal information, but their optimization complexity also rises polynomially.
>
> The other important factor is the **dataset**. OVIS, compared to YoutubeVIS 2019, has more occluded and crowded scenes (5.8 objects per video for OVIS while 1.7 for YoutubeVIS 2019). Thus, the complexity of OVIS for each frame is three times that of the YoutubeVIS 2019. In order to further reduce the complexity for better optimization, dividing into smaller sub-clips can accelerate the optimization. Also, keeping the size of two is able to maintain the temporal information.

---

> > ### Author Response · Authors · 2024-08-11
> >
> > Dear Reviewer MsYp,
> >
> > We sincerely appreciate your dedicated time and effort in reviewing our work. Your valuable comments and feedback are crucial for improving the quality of our research.
> >
> > We have carefully considered your concerns and provided corresponding responses and updated results. We believe we have addressed the key issues you raised, and we welcome further discussion to ensure we have fully covered your concerns. Please let us know if any remaining unclear parts or areas require additional clarification. We are committed to providing a comprehensive response and are open to any additional feedback you may have. Your input is invaluable in helping us strengthen our work.
> >
> > Thank you again for your thorough review. We look forward to continuing our productive discussion.

---

> ### Comment · Reviewer_MsYp · 2024-08-13
>
> Thanks to the author for the response. All my doubts have been answered in the author's rebuttal. I agree with the phenomena observed on OVIS and hope that there will be more detailed explanations about Synchronized Embedding Optimization in the revision.

---

### Official Review · Reviewer_3JCg · 2024-07-16

**Soundness:** 3
**Presentation:** 3
**Contribution:** 3
**Rating:** 7
**Confidence:** 3

**Summary:**

This paper proposes SyncVIS, an approach for Video Instance Segmentation (VIS), which tries to jointly model frame-level and video-level embeddings thus can capture both semantics and movement of instances. The new architecture design is intuitive and generic enough to be applied to various VIS models. Experiments are done on Youtube-VIS 2019, 2021, 2022, and OVIS which show that SyncVIS achieves state-of-the-art results on these benchmarks. Ablations are thorough and enough to understand the proposed approach. Written presentation is clear and easy to read.

**Strengths:**

- The newly proposed architecture design is intuitive and effective.
- The new approach can be applied to most of existing VIS architectures and gives consistent improvements.
- SyncVIS achieves state-of-the-art performance on current benchmarks.
- Ablation experiments are solid and provide good insights about the newly proposed method.

**Weaknesses:**

- The paper may be benefit from having from qualitative results to highly what SyncVIS can improve from the base models which were trained on asynchronous manner in the main text if space allows.

**Questions:**

- Why the results of Mask2Former-VIS on OVIS dataset in Table 9 are too low?

**Limitations:**

The author(s) have some statements about their method's limitation and further provide more details in supplementary.

---

> ### Author Rebuttal · Authors · 2024-08-07
>
> In the beginning, we want to thank you for the detailed, insightful, and constructive comments.
>
> ### Qualitative results
> In the Appendix, we provide many illustrations comparing previous base models with our SyncVIS. We select some cases under different scenarios, which include settings with multiple similar instances, settings with reappearance of instance, settings with different poses of instance, and settings with long video (Fig. 4 - Fig. 7). In the Fig. 8 of the Appendix, we provide visualizations of implementing different designs of embeddings, which further illustrate our synchronous design is capable of better segmenting and tracking multiple similar instances, while the asynchronous design ends in ignoring some instances and segmenting objects incompletely.
>
> ### Mask2Former-VIS on OVIS dataset performance
>
> - **Complexity of OVIS**
>
> OVIS is known for having more occluded and crowded scenes than the YoutubeVIS dataset (5.8 objects per video for OVIS while 1.7 for YoutubeVIS 2019), and thus poses more challenges to the model's temporal modeling capacity on multiple instances.
> - **Mask2Former-VIS**
>
> Mask2Former-VIS, on the other hand, is a typical **offline** VIS method that models the whole video sequence with only video-level embeddings, which is extremely insufficient for handling occluded and crowded scenes. Even if we increase the input frame number to bring more temporal information, the modeling is still harder for Mask2Former-VIS. Even though we experimented with different hyperparameter settings for OVIS scenario, the results are still unsatisfying due to the innate restrictions of Mask2Former-VIS. The following-up work of Mask2Former-VIS, such as Seqformer, also suffers from the typical asynchronous design of offline methods, and its performance is also below 15 AP.

---

### Decision · Program_Chairs · 2024-09-25

**Decision:**

Accept (poster)

**Comment:**

All reviewers are inclined towards acceptance, with scores ranging from 5 to 7. They have highlighted several areas for improvement that the authors should address carefully in the final version. These include adding clarifications and detailed explanations and all suggested additional experiments, and enhancing the presentation and organization of the paper to minimize confusion.